# Differential antibody production by symptomatology in SARS-CoV-2 convalescent individuals

**Sharada Saraf[1], Xianming Zhu[2], Ruchee Shrestha[2], Tania S. Bonny[2¤], Owen R. Baker[3], Evan J. Beck[1], Reinaldo E. Fernandez[3], Yolanda Eby[2], Olivia Akinde[2], Jessica E. Ruff[2], Patrizio Caturegli[2], Andrew D. Redd[1,3], Evan M. Bloch****[2], Thomas C. Quinn[1,2,3], Aaron A. R. Tobian[2,3,4], Oliver Laeyendecker****[1,3]***

**1** Division of Intramural Research, National Institute of Allergy and Infectious Diseases, National Institutes of Health, Bethesda, Maryland, United States of America, **2** Department of Pathology, Johns Hopkins School of Medicine, Baltimore, Maryland, United States of America, **3** Division of Infectious Diseases, Department of Medicine, Johns Hopkins School of Medicine, Baltimore, Maryland, United States of America, **4** Department of Epidemiology, Johns Hopkins Bloomberg School of Public Health, Baltimore, Maryland, United States of America

¤ Current address: U.S. Food and Drug Administration, Silver Spring, Maryland, United States of America
* olaeyen1@jhmi.edu

**Data Availability Statement:** All relevant data are within the manuscript and its Supporting Information files.

**Funding:** This work was supported in part by the Division of Intramural Research, National Institute

## Abstract

The association between COVID-19 symptoms and antibody responses against SARS-CoV-2 is poorly characterized. We analyzed antibody levels in individuals with known SARS-CoV-2 infection to identify potential antibody-symptom associations. Convalescent plasma from 216 SARS-CoV-2 RNA+ individuals with symptomatology information were tested for the presence of IgG to the spike S1 subunit (Euroimmun ELISA), IgG to receptor binding domain (RBD, CoronaCHEK rapid test), and for IgG, IgA, and IgM to nucleocapsid (N, Bio-Rad ELISA). Logistic regression was used to estimate the odds of having a COVID-19 symptom from the antibody response, adjusting for sex and age. Cough strongly associated with antibodies against S1 (adjusted odds ratio [aOR] = 5.33; 95% CI from 1.51 to 18.86) and RBD (aOR = 4.36; CI 1.49, 12.78). In contrast, sore throat significantly associated with the absence of antibodies to S1 and N (aOR = 0.25; CI 0.08, 0.80 and aOR = 0.31; 0.11, 0.91). Similarly, lack of symptoms associated with the absence of antibodies to N and RBD (aOR = 0.16; CI 0.03, 0.97 and aOR = 0.16; CI 0.03, 1.01). Cough appeared to be correlated with a seropositive result, suggesting that SARS-CoV-2 infected individuals exhibiting lower respiratory symptoms generate a robust antibody response. Conversely, those without symptoms or limited to a sore throat while infected with SARS-CoV-2 were likely to lack a detectable antibody response. These findings strongly support the notion that severity of infection correlates with robust antibody response.

## Introduction

The ongoing COVID-19 pandemic has challenged health care systems globally and necessitated rapid deployment of treatments and vaccines. SARS-CoV-2 infection, the causative agent

of Allergy and Infectious Diseases (NIAID) (OL, ADR, TCQ), as well as extramural support from NIAID (R01AI120938, R01AI120938S1 and R01AI128779 to AART; funders had no role in study design, data analysis, decision to publish, or preparation of the manuscript.

**Competing interests:** The authors have declared that no competing interests exist.

of COVID-19, elicits a broad range of symptoms: fever, cough, shortness of breath, and myalgia are the most reported symptoms among critically ill patients [1]. Antibody levels serve as a potential correlate of protection against COVID-19; individuals who test positive for anti-spike and anti-nucleocapsid IgG antibodies have demonstrated a substantially reduced risk of SARS-CoV-2 reinfection [2]. Moreover, high vaccine-induced antibody responses are associated with lower risk of symptomatic COVID-19 [3].

Previous studies have observed higher prevalence of seroconversion among severely ill individuals versus those with asymptomatic or mild disease [4]. Additionally, studies have shown that males, older individuals, and those previously hospitalized with symptoms generate strong antibody responses [5]. SARS-CoV-2 antibody levels have been demonstrated to positively correlate with the severity of COVID-19; however, the immune responses of individuals experiencing milder disease remain poorly characterized [6–8]. Investigating possible correlations with symptomatology can add more nuance to characterizing population level immunity or seroprevalence in a certain population, thus informing future public health interventions [7,9]. Furthermore, these data may help inform whether previously infected individuals have a higher chance of re-infection depending on their symptom presentation during their disease course, which can better characterize the urgency of vaccination in these individuals [10,11].

We investigated whether certain symptoms are predictive of a stronger antibody response by analyzing the antibody levels of individuals with known SARS-CoV-2 infection for associations between antibody response and reported symptoms. Samples from individuals who recovered from SARS-CoV-2 infection were tested for the presence of IgG antibodies to spike (S1), IgG antibodies to the receptor binding domain (RBD), and total antibodies to nucleocapsid (N).

## Materials and methods

### Study participants

This study used stored samples and data from studies that were approved by The Johns Hopkins University School of Medicine Institutional Review Board. All study participants provided written informed consent and were de-identified prior to laboratory testing.

To assess the antibody levels of SARS-CoV-2 infected individuals, samples from 216 participants from the Baltimore/Washington DC area who were screened to donate COVID-19 convalescent plasma (CCP) and had accompanying symptom data from April 2020-January 2021 were evaluated [5,12,13]. All were at least 18 years old and met the eligibility criteria for blood donation. Participants were engaged in a larger clinical trial investigating the use of convalescent plasma for prevention and treatment of COVID-19; recruitment efforts included community referral, employee referral, and existing blood donation registries. These were targeted at individuals in the Baltimore/Washington DC area who had a positive test for COVID-19 and were symptom-free at the time of screening. 22.6% of participants reported being medical professionals. The exclusion criteria included receipt of any experimental COVID-19 medication or vaccine as well as antiplatelet agents, anticoagulants, isotretinoin, finasteride, dutasteride, vismodegib, teriflunomide, acitretin, etretinate, and hepatitis B immune globin.

### Ascertainment of the symptomatology

As a part of a phone screening, participants were asked by a study team member if they were hospitalized and/or experienced any symptoms during their illness and, if so, to list their symptoms. Participant answers were then recorded by the screener according to 17 standard categories: no symptoms, fever, cough, chills, shortness of breath, diarrhea, fatigue, anosmia, dysgeusia, sore throat, headache, muscle ache, runny nose, stuffy nose, nausea, vomiting, or other.

## Laboratory methods

Plasma was separated from whole blood within 12 hours of collection and stored at −80˚C until further testing. Samples were analyzed using three commercially available serologic assays: Euroimmun Anti-SARS-CoV-2 ELISA (Mountain Lakes, NJ), the CoronaCHEK™ COVID-19 IgG/IgM Rapid Test Cassette (Hangzhou Biotest Biotech Co Ltd), and the Bio-Rad Platelia SARS-CoV-2 Total Antibody ELISA (Marnes-la-Coquette, France). The Euroimmun ELISA measures IgG responses to the SARS-CoV-2 S1 protein; the manufacturer reported an estimated sensitivity of 90% (95% CI 74%, 97%) and specificity of 100% (95% CI 95%, 100%) [14]. The CoronaCHEK rapid test measures IgG responses to the SARS-CoV-2 RBD, with a reported sensitivity of 97% (95% CI 83%, 99%) and specificity of 98% (95% CI 91%, 99%) [14,15]. The Bio-Rad ELISA measures total antibody response to the SARS-CoV-2 N, with a reported plasma sensitivity of 96% (95% CI 79%, 99%) and specificity of 100% (95% CI 99%, 100%) [16].

Thirty-five cytokine and chemokine analytes in plasma were assessed using a multi-array electrochemiluminescence detection technology (MesoScale Discovery, Gaithersburg, MD) as previously described [17]. Analytes with ≥80% overall detectability were evaluated for cytokine level differences between symptom groups and included Eotaxin, Eotaxin-3, IFN-γ, IL-12/IL23p40, IL-15, IL-16, IL-17A, IL-18, IL-1RA, IL-6, IL-7, IL-8, IP-10, MCP-1, MCP-4, MDC, MIP-1β, TARC, TNF-α, and VEGF-A. Analytes with <80% overall detectability were evaluated for percent detectability differences between symptom groups and included IL-12p70, IL-13, IL-1β, IL-2, IL-4, G-CSF, IFN-α2a, IL-21, IL-33, IL-8(HA), MIP-1α, GM-CSF, IL-1α, IL-5, and TNF-β. All assays were performed according to the manufacturer's protocols.

## Statistical analysis

Binomial logistic regressions were performed to calculate odds ratios [OR] for associations between serological results and reported symptoms. Adjusted odds ratios [aOR] were calculated for all symptoms. Based on previous studies linking sex and age to antibody reactivity, these were considered to be confounding variables and therefore included in adjusted models [5]. Adjusted odds ratios with a $p<0.05$ were considered significant. All analysis were performed in STATA v.14.2 (College Station, TX).

## Results

Participants were a median age of 49 years (IQR 37–58) at the time of sample collection. This subject pool was 81.9% White, 9.7% Black, 4.2% Asian, and 4.2% mixed/other/unknown (Table 1). A median of 49 days (IQR 40–64) had elapsed since participants had a confirmed SARS-CoV-2 diagnosis via detectable RNA.

Of the 17 different categories, the most frequently reported were fatigue (53%), fever (50%), and cough (50%) (Fig 1). Headache (44%), muscle ache (43%), loss of smell (38%), altered taste (33%), short breath (26%), stuffy nose (25%), and sore throat (20%) were also commonly reported. Chills (16%), diarrhea (15%), nausea (9%), runny nose (8%), no symptoms (5%), and vomiting (3%) were the least recorded categories. Hospitalization occurred in 7% of all participants. Individuals reporting fatigue also commonly reported headache (29%), fever (26%), cough (26%), and muscle ache (26%).

For each of the three serologic assays, >83% of all samples had a positive result. All individuals who were hospitalized had reactive plasma to the Euroimmun ELISA, CoronaCHECK (IgG) rapid test, and Bio-Rad ELISA. For individuals reporting shortness of breath reactivity on Euroimmun, CoronaCHECK IgG, and Bio-Rad assays were positive on 93%, 91%, and 86% respectively (Fig 2). Other symptoms had similar consistency in reactivity, with fever and cough specifically demonstrating a similar range of percent reactivity (88–94%) across the

**Table 1. Demographic data of convalescent plasma donors.**

|  | All | Female | Male |
|---|---|---|---|
| Number of individuals | 216 | 137 | 79 |
| Median age (IQR) | 49 (37–58) | 49 (37–57) | 49 (38–61) |
| Age categories |  |  |  |
| 19–44 | 85 | 54 | 31 |
| 45–64 | 106 | 69 | 37 |
| 65+ | 25 | 14 | 11 |
| Race/ethnicity |  |  |  |
| White | 177 | 111 | 66 |
| Black | 21 | 15 | 6 |
| Asian | 9 | 7 | 2 |
| Other | 9 | 4 | 5 |
| Median days post PCR+ blood collection (IQR) | 49 (40–64) | 54 (42–75) | 43 (38–58) |

Abbreviations: IQR, inter quartile range.

three assays. Similarly, lack of reactivity to these two assays appeared to be consistent, with the exception of vomiting. Lack of symptoms (40–60%) and sore throat (73–75%) demonstrated relative stability across all three assays.

Signal to cut-off ratios (S/C) were generated for the Euroimmun and BioRad ELISAs. These results were stratified by five 5 symptom categories: cough, sore throat, no symptoms, and other symptoms (**Fig 3**). For the Bio-Rad assay, individuals reporting cough or other symptoms had the highest mean S/C ratio. Sore throat and no symptoms had the lowest mean S/C ratio. Similarly, the highest S/C ratios on the Euroimmun assay were generated by samples from individuals reporting cough, other symptoms, sore throat, and no symptoms.

Individuals reporting cough had the strongest association with a positive antibody response to S1 (aOR = 5.33; 95% CI 1.51, 18.86) and RBD (aOR = 4.36; CI 1.49, 12.78) though not to N

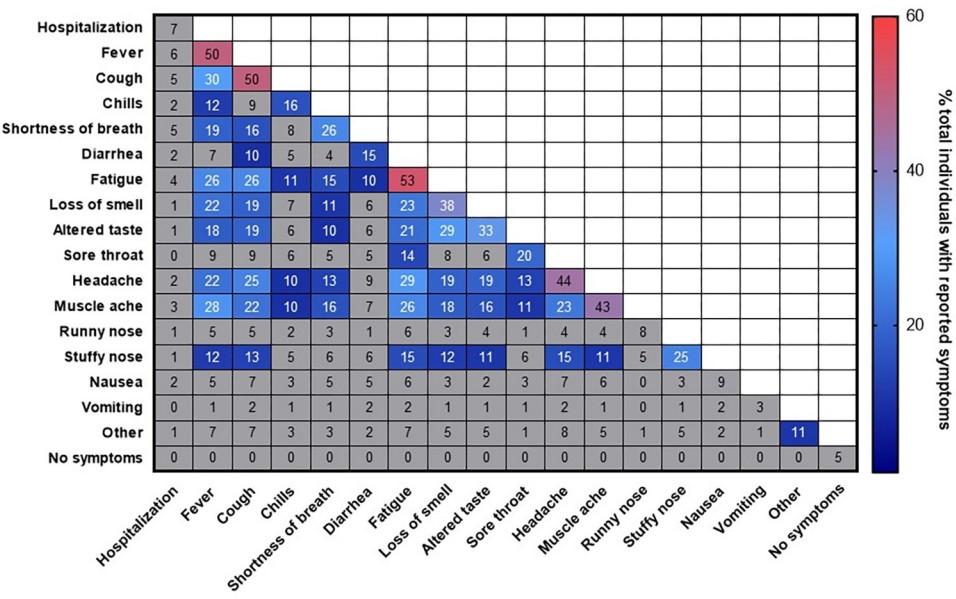

**Fig 1. Frequency and correlation of COVID-19 symptoms.** Percentage of individuals with the symptoms or pairs of symptoms are presented. Symptom or symptom pairs prevalent in >10% of individuals are colored.

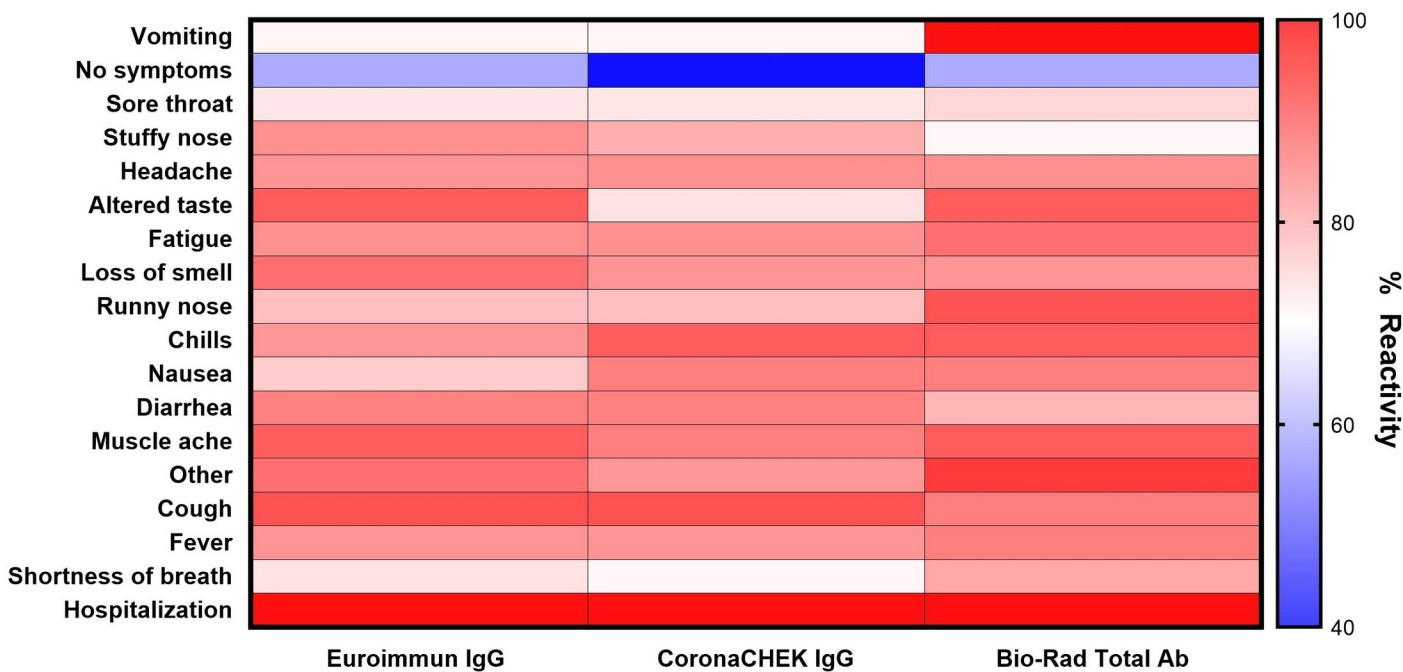

**Fig 2. Reactivity of antibody assays by presenting symptoms.** Percent reactivity was calculated by dividing the number of individuals with positive antibody results reporting the indicated symptom by the total number of individuals reporting the indicated symptom.

(**Table 2**). In contrast, sore throat was significantly associated with a lack of detectable antibody response to S1 and N (aOR = 0.25; CI 0.08, 0.80 and aOR = 0.31; 0.11, 0.91), respectively. Reporting a lack of symptoms was associated with a lack of antibody response to N (aOR = 0.16; CI 0.03, 0.97) and to RBD, though this association did not reach a statistical significance of <0.05 (aOR = 0.16; CI 0.03, 1.01). Individuals reporting diarrhea demonstrated decreased reactivity to N (aOR = 0.17; CI 0.05, 0.62). Notably, aORs and confidence intervals did not significantly attenuate after adjustment across assays for cough, sore throat, or no symptoms.

To further investigate the seronegativity of individuals reporting sore throat, individuals were grouped into three categories: whether they reported no symptoms, reported symptoms other than sore throat, or reported sore throat. Individuals who reported a sore throat and cough as co-symptoms were placed in the second category. Among the analytes with ≥80% overall detectability, the median cytokine levels were not significantly higher among convalescent individuals who were symptomatic, asymptomatic, or reporting sore throat (**Fig 4**). Among the analytes with <80% overall detectability, the percent detectability analytes in individuals reporting sore throat (no cough) versus other symptoms were not significantly different (**Fig 5**).

## Discussion

This study demonstrates associations between symptom presentation and antibody assay reactivity. Assay reactivity appears to be consistent across symptoms, suggesting that antibodies produced by convalescent individuals share a similar responsiveness to different parts of the virus regardless of symptom presentation. In addition to hospitalization and male sex, reporting cough appeared to be predictive of a seropositive result, suggesting that these individuals may be more likely to generate a robust antibody response. Conversely, sore throat and no symptoms were associated with a seronegative result.

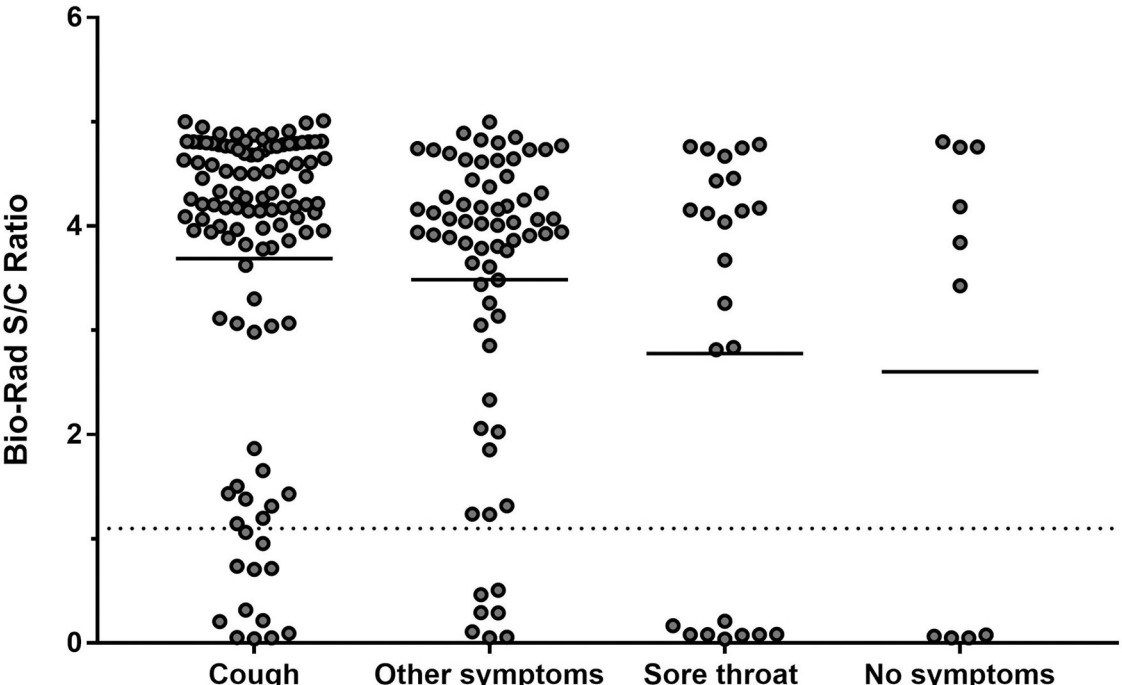

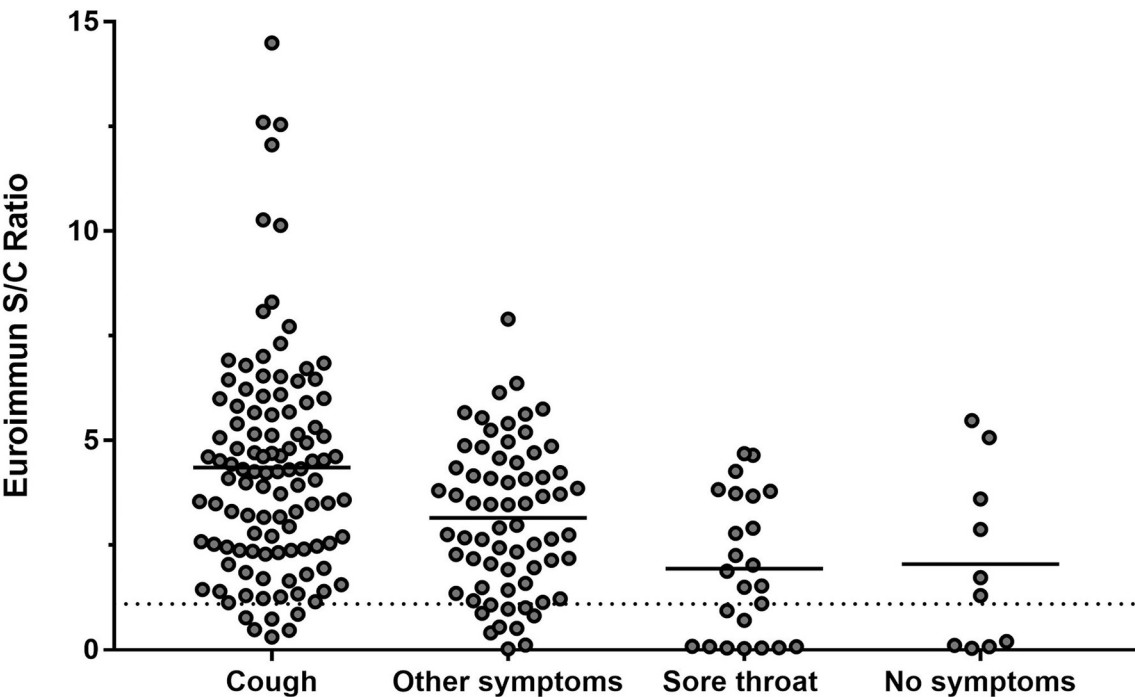

**Fig 3. Antibody reactivity to nucleocapsid protein as measured by Bio-Rad ELISA and S1 protein as measured by Euroimmun ELISA stratified by symptom category.** Solid horizontal lines represent the mean S/C ratio for the indicated symptom group. Dashed horizontal line represents the positive result threshold for the indicated assay.

**Table 2. The association between symptoms and antibody reactivity to S1, RBD and N proteins of SARS-CoV-2 among infected individuals.**

| Variable[1] | Euroimmun IgG S1 Positive Result[2] | | CoronaCHEK RBD Positive Result[2] | | BioRad Total Ab N Positive Result[2] | |
|---|---|---|---|---|---|---|
| | Crude Odds Ratio (95% CI) | Adjusted Odds Ratio (95% CI)[3] | Crude Odds Ratio (95% CI) | Adjusted Odds Ratio (95% CI)[3] | Crude Odds Ratio (95% CI) | Adjusted Odds Ratio (95% CI)[3] |
| Cough (n = 110) | 5.82 (2.12, 16.0) ‡ | 5.33 (1.51, 18.86) ‡ | 5.82 (2.19, 12.7) ‡ | 4.36 (1.49, 12.78) ‡ | 1.97 (0.89, 4.36) | 1.51 (0.55, 4.13) |
| Altered taste (n = 72) | 1.59 (0.64, 3.93) | 0.68 (0.14, 3.26) | 3.52 (1.31, 9.53) † | 3.48 (0.82, 14.82) | 1.44 (0.61, 3.42) | 0.75 (0.18, 3.17) |
| Sore throat (n = 45) | 0.28 (0.12, 0.66) ‡ | 0.25 (0.08, 0.80) † | 0.43 (0.19, 0.94) † | 0.48 (0.16, 1.38) | 0.39 (0.17, 0.87) † | 0.31 (0.11, 0.91) † |
| Muscle ache (n = 92) | 1.67 (0.72, 3.88) | 1.87 (0.63, 5.49) | 2.45 (1.09, 5.51) † | 2.25 (0.83, 6.09) | 1.58 (0.70, 3.55) | 2.23 (0.79, 6.26) |
| Diarrhea (n = 33) | 0.81 (0.28, 2.29) | 0.33 (0.08, 1.39) | 1.10 (0.39, 3.07) | 0.77 (0.19, 2.67) | 0.53 (0.21, 1.37) | 0.17 (0.05, 0.62) ‡ |
| No symptoms (n = 10) | 0.20 (0.05, 0.75) † | 0.24 (0.04, 1.49) | 0.11 (0.03, 0.41) ‡ | 0.16 (0.03, 1.01) | 0.22 (0.06–0.82) † | 0.16 (0.03, 0.97) † |

[1]Symptoms with no significant association with antibody reactivity found: Fatigue (n = 115), fever (n = 111), headache (n = 95), anosmia (n = 83), shortness of breath (n = 57), chills (n = 35), nausea (n = 20), stuffy nose (n = 39), runny nose (n = 18), vomiting (n = 7), other (n = 23).

[2]Abbreviation: S1, spike protein subunit 1; RBD, receptor binding domain; N, nucleocapsid; CI, confidence interval; n, number.

[3]Variables in the adjusted model included sex, age, and all symptoms.

† p < 0.05

‡ p < 0.01.

Previous studies have demonstrated higher antibody titers in individuals exhibiting more symptomatic disease [4,18]. Our results are complementary to these findings, given that asymptomatic convalescent individuals were significantly associated with a seronegative result. Strikingly, our study demonstrates the single symptom of sore throat being associated with a seronegative result. This finding has not been demonstrated in prior studies investigating COVID-19 symptoms and antibody reactivity. However, the lower and upper respiratory tract has shown to differ in their mechanisms of immunity, with sore throat being a presenting symptom of an upper respiratory tract infection [19–21].

Studies regarding influenza have demonstrated robust IgG responses to be more indicative of a lower respiratory tract infection [22,23]. Moreover, others have suggested that the progress

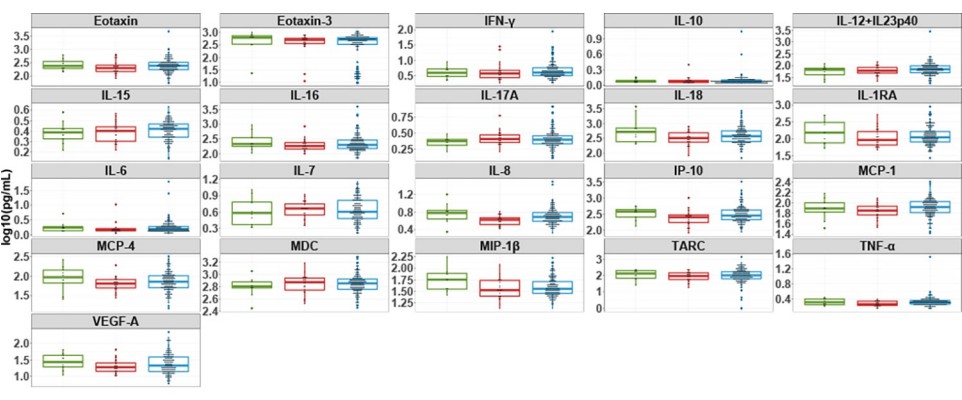

**Fig 4. Cytokine and chemokine levels by symptom group.** Log-transformed concentrations of cytokine and chemokines with ≥80% detectability in the overall sample are shown. The median and inter-quartile range as well as all data points are presented. Individuals reporting no symptoms (n = 10), sore throat and no cough (n = 24), and all other symptoms (n = 182) are shown in green, red, and blue, respectively.

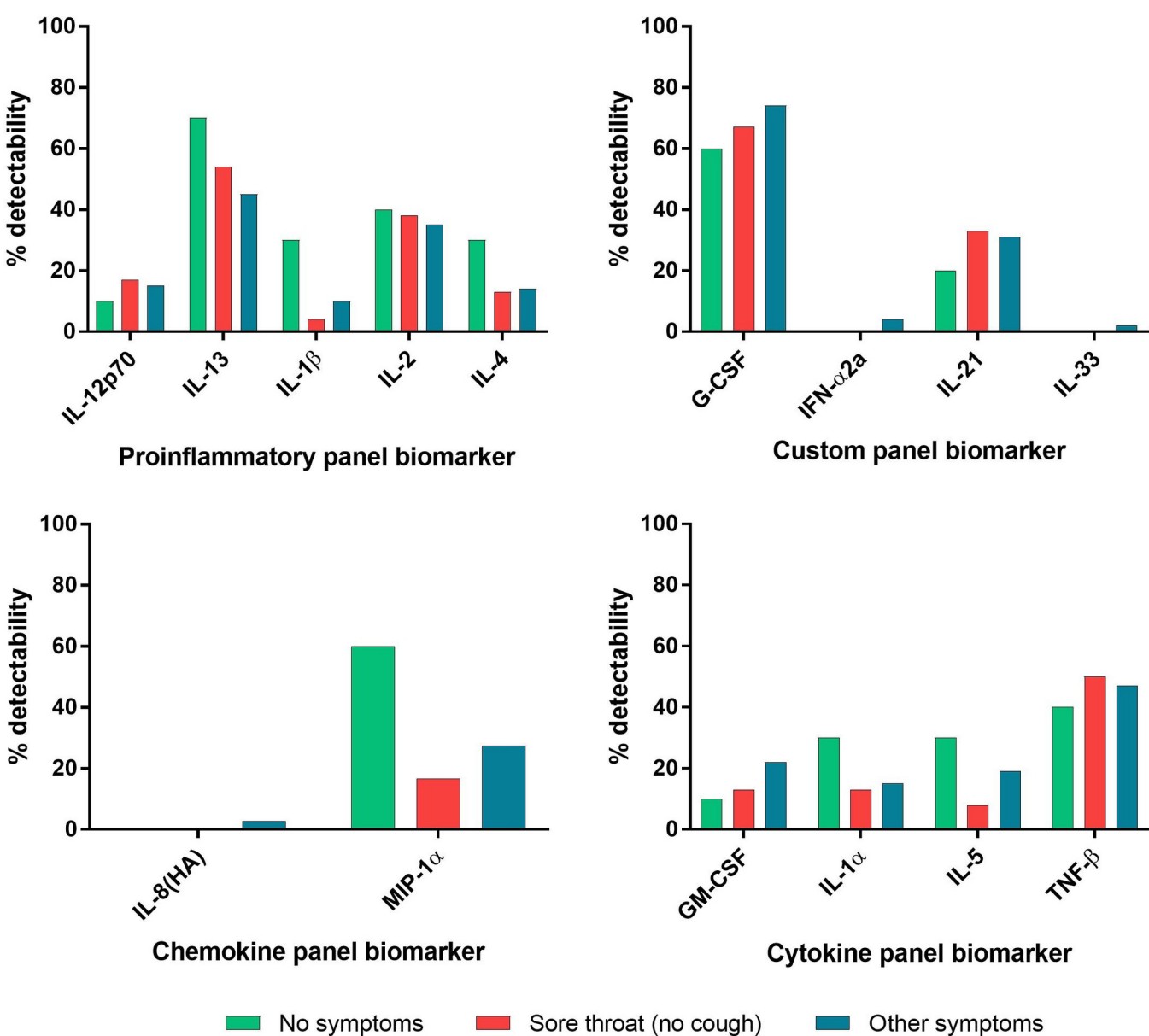

**Fig 5. Detectability of cytokine and chemokine markers by response panel and symptom group.** Cytokine and chemokine analytes with <80% detectability in the overall participant group are shown according to MSD panel. Individuals reporting no symptoms (n = 10), sore throat and no cough (n = 24), and all other symptoms (n = 182) are shown in green, red, and blue, respectively.

of COVID-19 disease, when confined to the upper respiratory tract, typically appears to resolve with minimal to no symptoms [24]. This literature may serve as a potential explanation as to why convalescent individuals in this study reporting no symptoms and sore throat generate fewer IgG antibodies. Alternatively, a strong innate immune response may serve to effectively combat the virus in these convalescent individuals, thus not necessitating a robust antibody response; Xu et al. measured cytokine, chemokine, and growth factor levels in COVID-19 patients with varying levels of disease severity and found that PDGF-BB, CCL5/RANTES, IL-9, TNF-β, and CCL4/MIP-1β are upregulated to higher levels in mild than severe and/or fatal COVID-19 patients [25]. However, our cytokine and chemokine data, which included TNF-β

and MIP-1β, do not demonstrate any significant differences among symptom groups and therefore do not support this rationalization. Given that a median of 30 days had passed since symptom resolution in this subject pool at the time of blood collection, it is possible that cytokine and chemokine levels may have declined to their basal levels [26].

Our study had several limitations. First, capture of clinical symptoms was based on self-reporting rather than review of the patients' medical records. Individuals reporting a certain symptom may have experienced a more severe presentation than others reporting the same symptom, which was not captured by this dataset and may have influenced antibody production. Second, samples were obtained a median of 49 days after participants had PCR positive results and 30 days post symptom resolution. Antibody levels may have declined at the time of sample collection; furthermore, samples were collected at only one timepoint and inferences about persistently high titers of antibodies based on symptom cannot be made. Moreover, current literature strongly suggests that neutralizing antibody levels are associated with disease severity; individuals with severe COVID-19 have been shown to develop higher neutralizing titers compared to patients with mild disease [27]. Neutralizing antibody assay result may have an association with symptom categories, but this was assay was not available to us and therefore not investigated by this study.

In this cohort of known SARS-CoV-2 infected individuals, we found a strong association between cough and antibody response. Conversely, a sore throat was strongly associated with a lack of antibody response to SARS-CoV-2 infection. Future studies could test for IgA levels in nasal or throat samples to evaluate whether a robust mucosal IgA response is associated with certain clinical presentations of COVID-19. Immune factors other than IgG antibodies, total antibodies, and our panel of human cytokines may also be evaluated to better characterize the immune responses generated by individuals exhibiting particular symptomatology.

## Supporting information

**S1 Data.**
(XLSX)

## Acknowledgments

The authors would like to thank the participants who donated their plasma and the study trial staff, without whom this investigation could not be conducted.

## Author Contributions

**Conceptualization:** Aaron A. R. Tobian, Oliver Laeyendecker.

**Data curation:** Xianming Zhu, Ruchee Shrestha, Yolanda Eby, Olivia Akinde, Jessica E. Ruff.

**Formal analysis:** Sharada Saraf, Xianming Zhu, Ruchee Shrestha, Tania S. Bonny, Owen R. Baker, Evan J. Beck, Reinaldo E. Fernandez, Evan M. Bloch.

**Funding acquisition:** Evan M. Bloch.

**Investigation:** Sharada Saraf, Tania S. Bonny, Owen R. Baker, Evan J. Beck, Olivia Akinde, Jessica E. Ruff, Patrizio Caturegli, Oliver Laeyendecker.

**Methodology:** Xianming Zhu, Patrizio Caturegli.

**Supervision:** Reinaldo E. Fernandez, Yolanda Eby, Andrew D. Redd, Aaron A. R. Tobian, Oliver Laeyendecker.

**Validation:** Sharada Saraf.

**Visualization:** Andrew D. Redd, Thomas C. Quinn, Oliver Laeyendecker.

**Writing – original draft:** Sharada Saraf, Reinaldo E. Fernandez, Andrew D. Redd, Thomas C. Quinn, Aaron A. R. Tobian, Oliver Laeyendecker.

**Writing – review & editing:** Xianming Zhu, Ruchee Shrestha, Tania S. Bonny, Owen R. Baker, Evan J. Beck, Yolanda Eby, Olivia Akinde, Jessica E. Ruff, Patrizio Caturegli, Andrew D. Redd, Evan M. Bloch, Thomas C. Quinn, Aaron A. R. Tobian, Oliver Laeyendecker.

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
