## [Decision Letter · Decision Letter 0]

24 Mar 2022

PONE-D-22-03797Differential antibody production by symptomatology in SARS-CoV-2 convalescent individualsPLOS ONE

Dear Dr. Laeyendecker,

Thank you for submitting your manuscript to PLOS ONE. After careful consideration, we feel that it has merit but does not fully meet PLOS ONE’s publication criteria as it currently stands. Therefore, we invite you to submit a revised version of the manuscript that addresses the points raised during the review process.

We look forward to receiving your revised manuscript.

Kind regards,

Han-Chung Wu, Ph.D.

Academic Editor

PLOS ONE

Journal Requirements:

2. Please note that in order to use the direct billing option the corresponding author must be affiliated with the chosen institute. Please either amend your manuscript to change the affiliation or corresponding author, or email us at plosone@plos.org with a request to remove this option.

Reviewers' comments:

Reviewer's Responses to Questions

**Comments to the Author**

1. Is the manuscript technically sound, and do the data support the conclusions?

Reviewer #1: Yes

Reviewer #2: Yes

2. Has the statistical analysis been performed appropriately and rigorously? 

Reviewer #1: Yes

Reviewer #2: Yes

3. Have the authors made all data underlying the findings in their manuscript fully available?

Reviewer #1: Yes

Reviewer #2: Yes

4. Is the manuscript presented in an intelligible fashion and written in standard English?

Reviewer #1: Yes

Reviewer #2: Yes

5. Review Comments to the Author

Reviewer #1: Dear Editor,

Thank you for the opportunity to revise the manuscript PONE-D-22-03797 “Differential antibody production by symptomatology in SARS-CoV-2 convalescent individuals”.

This manuscript reported results on the association between self-reported symptoms and immune response in a sample of COVID19 PCR confirmed cases. The study results may contribute to the understanding of the host-virus interplay in COVID19, a topic of public health interest. One of the added values of this study is the assessment of both antibody response and cytokine and chemokine levels.

The manuscript is well written even if the methods and discussion sections should be expanded to allow the assessment of the findings’ external validity of results and increase the value of the study.

Please find my comments below:

1. Methods: The main issue is related to the sample selection. The description of the selection process is not reported and it is not clear to what degree the study sample differs from the COVID19 population in the study area and timespan. A more detailed description of the selection process should be reported in the methods.

2. Methods: The technical performance reported by the manufacturers of the antibody test used in the study may help the reader in the interpretation of the study results.

3. Methods: Since COVID19 vaccines are available by December 2020, a comment on the assessment of vaccination status of included patients may be appropriate.

4. Methods/Discussion: Since plasma samples’ collection occurred in a non-negligible timespan, could you assess and/or comment the potential impact of differences in time from diagnosis on differences in antibody response.

5. Discussion: An interesting finding is the association between specific symptoms and the antibody response to specific antigens. Expanding the discussion on the potential biological reasons underling, these associations may add value to the study results.

6. Discussion: A comparison of existing evidence on cytokine and chemokine levels for disease severity may also add value to the manuscript since they role has been widely studied in COVID19.

7. Table 2: It is not clear the sentence “Variables in the adjusted model included sex, age, and all symptom predictor variables.” What variables are included among the “symptom predictor variables”?

Best regards

Reviewer #2: Previous studies have shown that the titers of serologic immunoglobulin against SARS-CoV-2 is positively related to the severity of COVID-19. However, the immune responses of individuals experiencing milder disease remain unclear. The results of this study may be complementary to these previous findings, given that asymptomatic convalescent individuals were significantly associated with a seronegative result. COVID-19 pandemic is still ongoing around the world. This study is helpful for us to understand the antibody level with symptoms more. Thus I recommend PLOS ONE should accept this manuscript after minor revision.

1. Here the authors have evaluated the titers of IgG against RBD. However, the level neutralizing antibodies have been demonstrated to correlate with the severity or prognosis of COVID-19. The authors should describe this critical issue in the discussion section.

2. Did 219 participants who donated CCP receive any COVID-19 vaccine or antiviral drug?

3. Lack of figure legend in Fig.4 and 5

6. PLOS authors have the option to publish the peer review history of their article (what does this mean?). If published, this will include your full peer review and any attached files.

Reviewer #1: **Yes: **Francesco Venturelli

Reviewer #2: No

---

## [Author Response · Author response to Decision Letter 0]

8 Apr 2022

RESPONSE TO REVIEWERS: 

Re: PONE-D-22-03797 (“Differential antibody production by symptomatology in SARS-CoV-2 convalescent individuals”)

Reviewer #1 (Major Comments for the Author):

1. This manuscript reported results on the association between self-reported symptoms and immune response in a sample of COVID19 PCR confirmed cases. The study results may contribute to the understanding of the host-virus interplay in COVID19, a topic of public health interest. One of the added values of this study is the assessment of both antibody response and cytokine and chemokine levels. The manuscript is well written even if the methods and discussion sections should be expanded to allow the assessment of the findings’ external validity of results and increase the value of the study.

RESPONSE: We thank the reviewer for their suggestions and appreciate their kind appraisal of the manuscript.

Reviewer #1 (Minor Comments for Author):

2. Methods: The main issue is related to the sample selection. The description of the selection process is not reported and it is not clear to what degree the study sample differs from the COVID19 population in the study area and timespan. A more detailed description of the selection process should be reported in the methods.

RESPONSE: A section further describing the selection process has been added to the methods.

Line 86-93: “Participants were engaged in a larger clinical trial investigating the use of convalescent plasma for prevention and treatment of COVID-19; recruitment efforts included community referral, employee referral, and existing blood donation registries. These were targeted at individuals in the Baltimore/Washington DC area who had a positive test for COVID-19 and were symptom-free at the time of screening. 22.6% of participants reported being medical professionals. The exclusion criteria included receipt of any experimental COVID-19 medication or vaccine as well as antiplatelet agents, anticoagulants, isotretinoin, finasteride, dutasteride, vismodegib, teriflunomide, acitretin, etretinate, and hepatitis B immune globin.”

3. Methods: The technical performance reported by the manufacturers of the antibody test used in the study may help the reader in the interpretation of the study results.

RESPONSE: A section describing the technical performance of the Euroimmun, CoronaCHEK, and Bio-Rad assays has been added to the methods section. Reference #16 has been updated with a source that includes Bio-Rad’s report on assay performance.

Lines 107-113: “The Euroimmun ELISA measures IgG responses to the SARS-CoV-2 S1 protein; the manufacturer reported an estimated sensitivity of 90% (95% CI 74%, 97%) and specificity of 100% (95% CI 95%, 100%).14 The CoronaCHEK rapid test measures IgG responses to the SARS-CoV-2 RBD, with a reported sensitivity of 97% (95% CI 83%, 99%) and specificity of 98% (95% CI 91%, 99%).15 The Bio-Rad ELISA measures total antibody response to the SARS-CoV-2 N, with a reported plasma sensitivity of 96% (95% CI 79%, 99%) and specificity of 100% (95% CI 99%, 100%).16”

4. Methods: Since COVID19 vaccines are available by December 2020, a comment on the assessment of vaccination status of included patients may be appropriate.

RESPONSE: A line clarifying the negative vaccination status of included participants has been added.

Line 91-92: “The exclusion criteria included receipt of any experimental COVID-19 medication or vaccine”

5. Methods/Discussion: Since plasma samples’ collection occurred in a non-negligible timespan, could you assess and/or comment the potential impact of differences in time from diagnosis on differences in antibody response.

RESPONSE: In our dataset, we found that the relationship between time from diagnosis on antibody test result to be insignificant (P> 0.10 for all . We reran the regression analysis and that the time from PCR diagnosis not to impact the odds ratios of the factors which were associated with the presence or absence of antibodies to SARS-CoV-2 in our population. While there is a great deal of evidence for the declining of antibody titer which occurs after several months from viral clearance, we found for our limited data set that the presence or absence of particular symptoms had a greater influence on the detectability of these antibodies than duration from diagnosis by positive PCR test. 

6. Discussion: An interesting finding is the association between specific symptoms and the antibody response to specific antigens. Expanding the discussion on the potential biological reasons underling, these associations may add value to the study results.

RESPONSE: While the authors agree that this is an interesting finding, existing literature does not provide basis for us to propose a potential biological reason underlying these associations; therefore, any reasons provided would be conjecture.

7. Discussion: A comparison of existing evidence on cytokine and chemokine levels for disease severity may also add value to the manuscript since they role has been widely studied in COVID19.

RESPONSE: This comparison has been added to the discussion section.

Lines 240-246: Xu et al. measured cytokine, chemokine, and growth factor levels in COVID-19 patients with varying levels of disease severity and found that PDGF-BB, CCL5/RANTES, IL-9, TNF-β, and CCL4/MIP-1β are upregulated to higher levels in mild than severe and/or fatal COVID-19 patients. However, our cytokine and chemokine data, which included TNF-β, and MIP-1β, do not demonstrate any significant differences among symptom groups and therefore do not support this rationalization.

8. Table 2: It is not clear the sentence “Variables in the adjusted model included sex, age, and all symptom predictor variables.” What variables are included among the “symptom predictor variables”?

RESPONSE: A sentence clarifying this has been added to Table 2. The symptom predictor variables are the 16 symptoms described in the methods section, which were included in order to minimize the effect of other symptom groups on the symptom of interest.

Table 2. Association between symptoms and antibody reactivity to S1, RBD and N proteins of SARS-CoV-2 among infected individuals

Variable1 Euroimmun IgG S1 Positive Result2 

 CoronaCHEK RBD Positive Result2 BioRad Total Ab N Positive Result2

 Crude Odds Ratio (95% CI) Adjusted Odds Ratio (95% CI)3 Crude Odds Ratio (95% CI) Adjusted Odds Ratio (95% CI)3 Crude Odds Ratio (95% CI) Adjusted Odds Ratio (95% CI)3

Cough (n=110) 5.82 (2.12, 16.0) ‡ 5.33 (1.51, 18.86) ‡ 5.82 (2.19, 12.7) ‡ 4.36 (1.49, 12.78) ‡ 1.97 (0.89, 4.36) 1.51 (0.55, 4.13)

Altered taste (n=72) 1.59 (0.64, 3.93) 0.68 (0.14, 3.26) 3.52 (1.31, 9.53) † 3.48 (0.82, 14.82) 1.44 (0.61, 3.42) 0.75 (0.18, 3.17)

Sore Throat (n=45) 0.28 (0.12, 0.66) ‡ 0.25 (0.08, 0.80) † 0.43 (0.19, 0.94) † 0.48 (0.16, 1.38) 0.39 (0.17, 0.87) † 0.31 (0.11, 0.91) †

Muscle ache (n=92) 1.67 (0.72, 3.88) 1.87 (0.63, 5.49) 2.45 (1.09, 5.51) † 2.25 (0.83, 6.09) 1.58 (0.70, 3.55) 2.23 (0.79, 6.26)

Diarrhea (n=33) 0.81 (0.28, 2.29) 0.33 (0.08, 1.39) 1.10 (0.39, 3.07) 0.77 (0.19, 2.67) 0.53 (0.21, 1.37) 0.17 (0.05, 0.62) ‡

Stuffy nose (n=39) 0.78 (0.29, 2.08) 1.55 (0.41, 5.84) 0.86 (0.35, 2.14) 0.83 (0.25, 2.77) 3.48 (0.79, 15.26) 5.07 (0.93, 27.71) †

No symptoms (n=10) 0.20 (0.05, 0.75)† 0.24 (0.04, 1.49) 0.11 (0.03, 0.41) ‡ 0.16 (0.03, 1.01) 0.22 (0.06-0.82) † 0.16 (0.03, 0.97) †

1Symptoms with no significant association with antibody reactivity found: fatigue (n=115), fever (n=111), headache (n=95), anosmia (n=83), shortness of breath (n=57), chills (n=35), nausea (n=20), runny nose (n=18), vomiting (n=7), other (n=23).

2Abbreviation: S1, spike protein subunit 1; RBD, receptor binding domain; N, nucleocapsid; CI, confidence interval; n, number.

3Variables in the adjusted model included sex, age, and all symptoms. 

† p< 0.05, ‡ p< 0.01.

Reviewer #2 (Major Comments for the Author):

9. Previous studies have shown that the titers of serologic immunoglobulin against SARS-CoV-2 is positively related to the severity of COVID-19. However, the immune responses of individuals experiencing milder disease remain unclear. The results of this study may be complementary to these previous findings, given that asymptomatic convalescent individuals were significantly associated with a seronegative result. COVID-19 pandemic is still ongoing around the world. This study is helpful for us to understand the antibody level with symptoms more. Thus I recommend PLOS ONE should accept this manuscript after minor revision.

RESPONSE: We appreciate the reviewer’s kind appraisal of the manuscript.

Reviewer #2 (Minor Comments for Author):

10. Here the authors have evaluated the titers of IgG against RBD. However, the level neutralizing antibodies have been demonstrated to correlate with the severity or prognosis of COVID-19. The authors should describe this critical issue in the discussion section.

RESPONSE: A section describing neutralizing antibody titer correlation with disease severity has been added.

Lines 256-261: Moreover, current literature strongly suggests that neutralizing antibody levels are associated with disease severity; individuals with severe COVID-19 have been shown to develop higher neutralizing titers compared to patients with mild disease.27 Neutralizing antibody assay result may have an association with symptom categories, but this was assay was not available to us and therefore not investigated by this study.

11. Did 219 participants who donated CCP receive any COVID-19 vaccine or antiviral drug?

RESPONSE: See response #4 above.

12. Lack of figure legend in Fig.4 and 5

RESPONSE: Figure legends have been added to Figs.4 and 5.

Lines 204-207: Log-transformed concentrations of cytokine and chemokines with ≥80% detectability in the overall sample are shown. The median and inter-quartile range as well as all data points are presented. Individuals reporting no symptoms (n=10), sore throat and no cough (n=24), and all other symptoms (n=182) are shown in green, red, and blue, respectively.

Lines 213-215: Cytokine and chemokine analytes with <80% detectability in the overall participant group are shown according to MSD panel. Individuals reporting no symptoms (n=10), sore throat and no cough (n=24), and all other symptoms (n=182) are shown in green, red, and blue, respectively.

OTHER CHANGES: 

13. Table 2 incorrectly shows “Stuffy nose” having a significant association with the BioRad antibody result. A corrected table has been included and the lines 190-191 have been amended to reflect this change.

Table 2. The association between symptoms and antibody reactivity to S1, RBD and N proteins of SARS-CoV-2 among infected individuals

Variable1 Euroimmun IgG S1 Positive Result2 

 CoronaCHEK RBD Positive Result2 BioRad Total Ab N Positive Result2

 Crude Odds Ratio (95% CI) Adjusted Odds Ratio (95% CI)3 Crude Odds Ratio (95% CI) Adjusted Odds Ratio (95% CI)3 Crude Odds Ratio (95% CI) Adjusted Odds Ratio (95% CI)3

Cough (n=110) 5.82 (2.12, 16.0) ‡ 5.33 (1.51, 18.86) ‡ 5.82 (2.19, 12.7) ‡ 4.36 (1.49, 12.78) ‡ 1.97 (0.89, 4.36) 1.51 (0.55, 4.13)

Altered taste (n=72) 1.59 (0.64, 3.93) 0.68 (0.14, 3.26) 3.52 (1.31, 9.53) † 3.48 (0.82, 14.82) 1.44 (0.61, 3.42) 0.75 (0.18, 3.17)

Sore Throat (n=45) 0.28 (0.12, 0.66) ‡ 0.25 (0.08, 0.80) † 0.43 (0.19, 0.94) † 0.48 (0.16, 1.38) 0.39 (0.17, 0.87) † 0.31 (0.11, 0.91) †

Muscle ache (n=92) 1.67 (0.72, 3.88) 1.87 (0.63, 5.49) 2.45 (1.09, 5.51) † 2.25 (0.83, 6.09) 1.58 (0.70, 3.55) 2.23 (0.79, 6.26)

Diarrhea (n=33) 0.81 (0.28, 2.29) 0.33 (0.08, 1.39) 1.10 (0.39, 3.07) 0.77 (0.19, 2.67) 0.53 (0.21, 1.37) 0.17 (0.05, 0.62) ‡

No symptoms (n=10) 0.20 (0.05, 0.75)† 0.24 (0.04, 1.49) 0.11 (0.03, 0.41) ‡ 0.16 (0.03, 1.01) 0.22 (0.06-0.82) † 0.16 (0.03, 0.97) †

1Symptoms with no significant association with antibody reactivity found: fatigue (n=115), fever (n=111), headache (n=95), anosmia (n=83), shortness of breath (n=57), chills (n=35), nausea (n=20), stuffy nose (n=39), runny nose (n=18), vomiting (n=7), other (n=23).

2Abbreviation: S1, spike protein subunit 1; RBD, receptor binding domain; N, nucleocapsid; CI, confidence interval; n, number.

3Variables in the adjusted model included sex, age, and all symptoms.

† p< 0.05, ‡ p< 0.01.

14. Greek letters were not included in the original submitted text for the relevant cytokines and chemokines. The manuscript has been amended reflect this change.

15. Figure 4 title has been corrected and amended to include a consistent color scheme. 

 16. Figure 5 title has been corrected and amended to include Greek letters and a consistent color scheme.

---

## [Decision Letter · Decision Letter 1]

18 Apr 2022

Differential antibody production by symptomatology in SARS-CoV-2 convalescent individuals

PONE-D-22-03797R1

Dear Dr. Laeyendecker,

We’re pleased to inform you that your manuscript has been judged scientifically suitable for publication and will be formally accepted for publication once it meets all outstanding technical requirements.

Kind regards,

Han-Chung Wu, Ph.D.

Academic Editor

PLOS ONE

Additional Editor Comments (optional):

Reviewers' comments:

Reviewer's Responses to Questions

**Comments to the Author**

1. If the authors have adequately addressed your comments raised in a previous round of review and you feel that this manuscript is now acceptable for publication, you may indicate that here to bypass the “Comments to the Author” section, enter your conflict of interest statement in the “Confidential to Editor” section, and submit your "Accept" recommendation.

Reviewer #1: (No Response)

Reviewer #2: All comments have been addressed

2. Is the manuscript technically sound, and do the data support the conclusions?

Reviewer #1: Yes

Reviewer #2: Yes

3. Has the statistical analysis been performed appropriately and rigorously? 

Reviewer #1: Yes

Reviewer #2: Yes

4. Have the authors made all data underlying the findings in their manuscript fully available?

Reviewer #1: Yes

Reviewer #2: Yes

5. Is the manuscript presented in an intelligible fashion and written in standard English?

Reviewer #1: Yes

Reviewer #2: Yes

6. Review Comments to the Author

Reviewer #1: Dear Editor,

the authors have addressed all provided comments.

I feel that the manuscript is now suitable for publication.

The only minor comment I have Is the need of a reference for the larger trial within which the authors recruited the samples included in the current study.

Best regards

Reviewer #2: (No Response)

7. PLOS authors have the option to publish the peer review history of their article (what does this mean?). If published, this will include your full peer review and any attached files.

Reviewer #1: **Yes: **Venturelli Francesco

Reviewer #2: No

---

## [Editor Report · Acceptance letter]

2 Jun 2022

PONE-D-22-03797R1 

Differential antibody production by symptomatology in SARS-CoV-2 convalescent individuals 

Dear Dr. Laeyendecker:

I'm pleased to inform you that your manuscript has been deemed suitable for publication in PLOS ONE. Congratulations! Your manuscript is now with our production department. 

Kind regards, 

on behalf of

Prof. Han-Chung Wu 

Academic Editor

PLOS ONE